# Mapping oxygen concentration in the awake mouse brain

Declan G Lyons[1,2], Alexandre Parpaleix[1,2], Morgane Roche[1,2], Serge Charpak[1,2]*

[1]Institut National de la Santé et de la Recherche Médicale, U1128, Paris, France; [2]Laboratory of Neurophysiology and New Microscopies, Université Paris Descartes, Paris, France

**Abstract** Although critical for brain function, the physiological values of cerebral oxygen concentration have remained elusive because high-resolution measurements have only been performed during anesthesia, which affects two major parameters modulating tissue oxygenation: neuronal activity and blood flow. Using measurements of capillary erythrocyte-associated transients, fluctuations of oxygen partial pressure ($Po_2$) associated with individual erythrocytes, to infer $Po_2$ in the nearby neuropil, we report the first non-invasive micron-scale mapping of cerebral $Po_2$ in awake, resting mice. Interstitial $Po_2$ has similar values in the olfactory bulb glomerular layer and the somatosensory cortex, whereas there are large capillary hematocrit and erythrocyte flux differences. Awake tissue $Po_2$ is about half that under isoflurane anesthesia, and within the cortex, vascular and interstitial $Po_2$ values display layer-specific differences which dramatically contrast with those recorded under anesthesia. Our findings emphasize the importance of measuring energy parameters non-invasively in physiological conditions to precisely quantify and model brain metabolism.

*For correspondence: serge. charpak@parisdescartes.fr

**Competing interests:** The authors declare that no competing interests exist.

## Introduction

To understand the relationship between brain oxygenation and diseases associated with hypoxia or ischemia, it is important to first determine what fixes the resting value of tissue $Po_2$, that is, the concentration of oxygen in the interstitium that bridges oxygen delivery from erythrocytes to oxygen consumption by mitochondria. Numerous methods have been used to monitor brain oxygenation, and the most spatially resolved approaches have long relied on fine Clark-type electrodes (for review see *Ndubuizu and LaManna, 2007*), which have been used to report resting-state $Po_2$ and local oxygen consumption in various brain regions (*Lecoq et al., 2009*; *Masamoto et al., 2003*; *Offenhauser et al., 2005*; *Thompson et al., 2003*) during neuronal activation. However, these electrodes are invasive, do not faithfully report $Po_2$ in vessels and cannot easily be used to determine $Po_2$ in physiological conditions, that is, in awake, unstressed animals, avoiding the use of anesthetics. As anesthetics affect resting and evoked neuronal and astrocyte activity, arterial blood pressure and cerebral blood flow, the physiological values of cerebral interstitial $Po_2$ and their relationship to blood flow parameters in capillaries remain unknown.

Recently, a two-photon phosphorescent probe PtP-C343 has been generated (*Finikova et al., 2007*, *2008*) and two-photon phosphorescence lifetime microscopy (2PLM) has been used to obtain depth-resolved, micron-scale measurements of $Po_2$ in the anesthetized rodent brain (*Devor et al., 2011*; *Lecoq et al., 2011*; *Parpaleix et al., 2013*; *Sakadzić et al., 2010*; *Sakadžić et al., 2014*). In addition, by detecting single red blood cells (RBCs) during $Po_2$ measurement, we demonstrated the possibility of simultaneously monitoring blood flow and $Po_2$ in capillaries (*Lecoq et al., 2011*) and of detecting erythrocyte-associated transients (EATs), $Po_2$ fluctuations associated with each individual erythrocyte flowing in capillaries, which were first reported in mesentery capillaries (*Golub and*

**eLife digest** Brain cells need a constant supply of oxygen to fuel their activities. This oxygen is delivered by the flow of blood through the vessels in the brain. If the blood flow to brain tissue is cut off as happens in stroke, or if an individual stops breathing, the brain becomes deprived of oxygen and brain cells will be damaged and die. To better understand how the brain works in health and disease, scientists need to learn how much oxygen the blood must deliver to the brain tissue to adequately support the activities of brain cells.

Many studies have measured oxygen levels in the brain. However, these studies have looked only roughly and taken measurements from large areas of the brain, or they have involved animals receiving anesthesia, which can alter blood flow and oxygen use in the brain. Recently, scientists discovered that they could measure oxygen concentration at high detail in the brain of anesthetized rodents with a specialized microscope, by using molecules that emit light at a rate that depends on the local oxygen concentration.

Now, Lyons et al. have shown that this same technique can be used in mice that are awake. First, a piece of the skull was replaced with glass to create a small transparent window. Then, the animals were allowed to recover for a few weeks, and were trained to get them used to how they would be handled during the experiments. After this period, the oxygen concentrations and blood flow in different parts of the mouse brains were measured in fine detail using the microscope while the animals were awake and relaxed.

The experiments showed that oxygen levels in awake resting mice are actually lower than in anesthetized mice, and that oxygen levels differ between different parts of the mouse brain. This first detailed look at oxygen levels in the brain of awake animals will likely lead to more studies. For example, future studies may look at how quickly the brain uses oxygen under normal conditions and what happens in the brain during disease.

*Pittman, 2005*). We showed that in olfactory bulb glomeruli of anesthetized mice, one parameter of EATs, the $Po_2$ in between two red blood cells ($Po_2InterRBC$), is at equilibrium with, and thus reports, the $Po_2$ in the nearby neuropil (*Parpaleix et al., 2013*). This result implied that measurements of $Po_2InterRBC$ could provide a powerful tool to non-invasively map local interstitial oxygen concentration in the brain of awake animals.

Here, we report that in both the olfactory bulb glomerular layer and the somatosensory cortex of unstressed, awake, resting mice, the interstitial $Po_2$ (equivalent to $Po_2InterRBC$) has the same mean value of ~23 mm Hg, spanning over a range of about 40 mm Hg. This contrasts with the large differences of capillary hematocrit and RBC flow values observed between the two brain regions. In addition, we find that in the cortex capillary and interstitial $Po_2$ values display layer-specific differences, being lower in layer I than in layer II/III or layer IV. We also find that hemoglobin in brain capillaries is highly saturated with oxygen. Finally, we show that in both brain regions, the interstitial $Po_2$ is much lower during wakefulness than under isoflurane anesthesia.

## Results

### Habituation of mice to head-restraint

To ensure that we measured $Po_2$ in real physiological conditions, that is, in awake, unstressed animals, each animal was habituated to all the conditions present during 2PLM $Po_2$ measurements for several weeks prior to the experiments (see methods for detailed training procedures). In brief, over the course of 2–3 days, each mouse was habituated to handling, and trained to run on a treadmill placed in its cage. A titanium bar was then surgically attached to the cranium and then a cranial window implanted over the region of interest, either the olfactory bulb or the somatosensory cortex. Then, over 2–4 weeks, the mouse was progressively habituated to being head-fixed, via the attached bar, in the dark, below the objective of the two-photon microscope, and with the limbs and body free to move on the treadmill. Habituation was achieved when the animal remained calm for periods >1 hr in the set-up with short bouts of running (~30 s). On the day of recording, the animal was

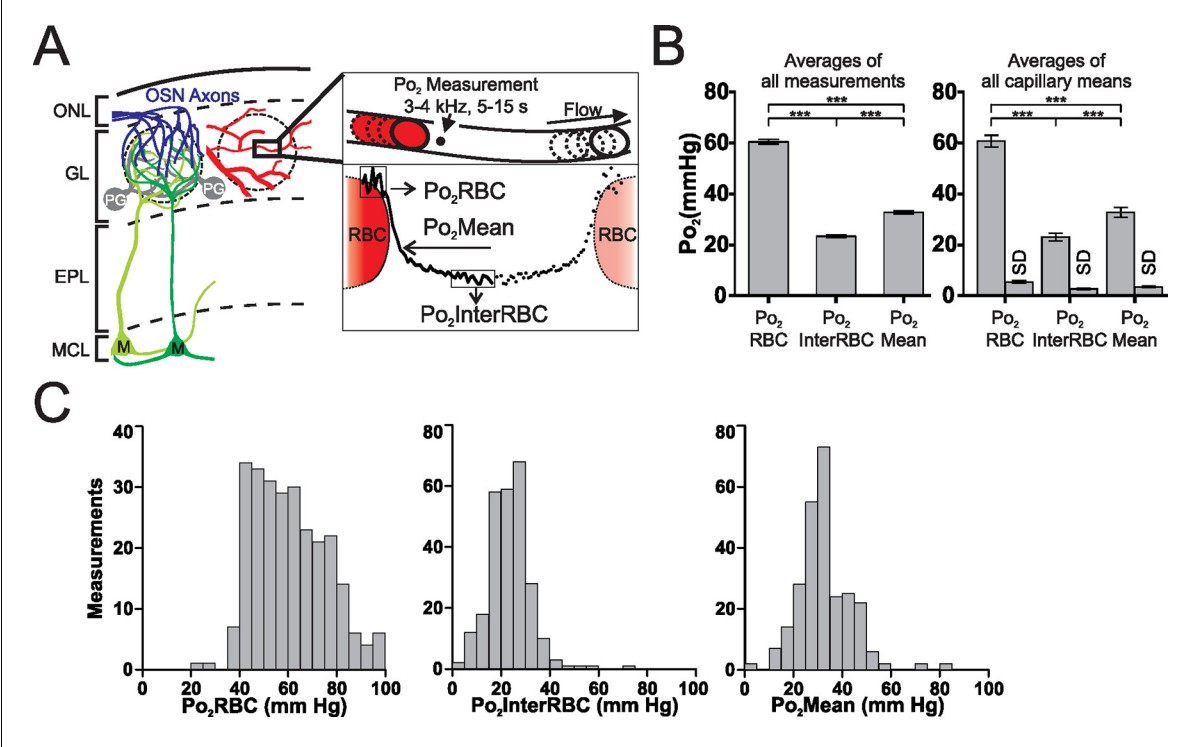

**Figure 1.** Erythrocyte-associated transients (EATs) in the olfactory bulb glomerular layer of the awake mouse. (**A**) Left panel, schematic diagram of the organisation of the olfactory bulb. OSN: olfactory sensory neuron, PG: periglomerular neuron, M: mitral cell, ONL: olfactory nerve layer, GL: glomerular layer, EPL: external plexiform layer, MCL: mitral cell layer. Right panel, top, schematic illustrating the 2PLM $Po_2$ measurement procedure in capillaries. Bottom, diagram showing $Po_2$ values extracted from EATs. The continuous trace represents the $Po_2$ profile relative to the RBC border in one selected capillary: $Po_2$ at the RBC border ($Po_2RBC$, in this case 47.2 mm Hg), $Po_2$ at distance from a RBC ($Po_2InterRBC$, in this case 8.6 mm Hg) which gives an estimate of $Po_2$ in the interstitium of the glomerular layer, and average $Po_2$ in the capillary ($Po_2Mean$, in this case 19 mm Hg). (**B**) Multiple (~4–8) measurements were made in each capillary. The $Po_2InterRBC$ is significantly lower than both $Po_2Mean$ and $Po_2RBC$, whether calculated on all measurements (the mean value is the average of all measurements pooled from all capillaries assessed, n = 262, left panel), or on the mean values from each capillary (the mean value is the average of the single mean values for each of the capillaries, n=38, right panel). SD (the average of the SD values for each capillary, presented as mean standard deviation ± SEM) illustrates the fluctuations of $Po_2$ values in each capillary across the multiple measurements. Data presented as mean ± SEM. *p<0.05, ***p<0.001, Kruskal-Wallis test with 2-tailed Dunn's multiple comparison post-hoc test. (**C**) Frequency distributions of all measurements of $Po_2RBC$ (left panel), $Po_2InterRBC$ (middle panel), and $Po_2Mean$ (right panel). 5 mm Hg bin width. For all plots n = 5 mice.

briefly anesthetized (2% isoflurane, <5 min) and the oxygen sensor PtP-C343 was injected intravenously. The animal was returned to its home cage, and after a delay of 90–120 min, $Po_2$ recordings sessions of 1–3 hr commenced. Each animal underwent 1–3 recording sessions per day over the course of 2–7 days, with breaks of at least several hours between each session. Note that similar $Po_2$ values were obtained from one day to the next and between sessions occurring the same day (without reinjection of PtP-C343).

## Characterization of EAT properties ($Po_2$ values) in the olfactory bulb of awake resting mice

Using our previous approach (**Lecoq et al., 2011**; **Parpaleix et al., 2013**), we characterized EATs in 38 capillaries (n = 5 animals) from the glomerular layer of awake resting mice (**Figure 1**). $Po_2$ measured at the RBC border ($Po_2RBC$) was significantly larger than at mid-distance between two RBCs ($Po_2InterRBC$). $Po_2Mean$, which was intermediate between these two values, is the average $Po_2$ measured in a capillary without taking into account the existence of EATs, and is the only capillary $Po_2$ value that has commonly been reported in the brain (**Sakadzić et al., 2010**; **Vovenko, 1999**). Several measurements were made in each capillary but the average values of $Po_2RBC$, $Po_2InterRBC$ and $Po_2Mean$ were similar whether calculated on all measurements or on all capillaries (262

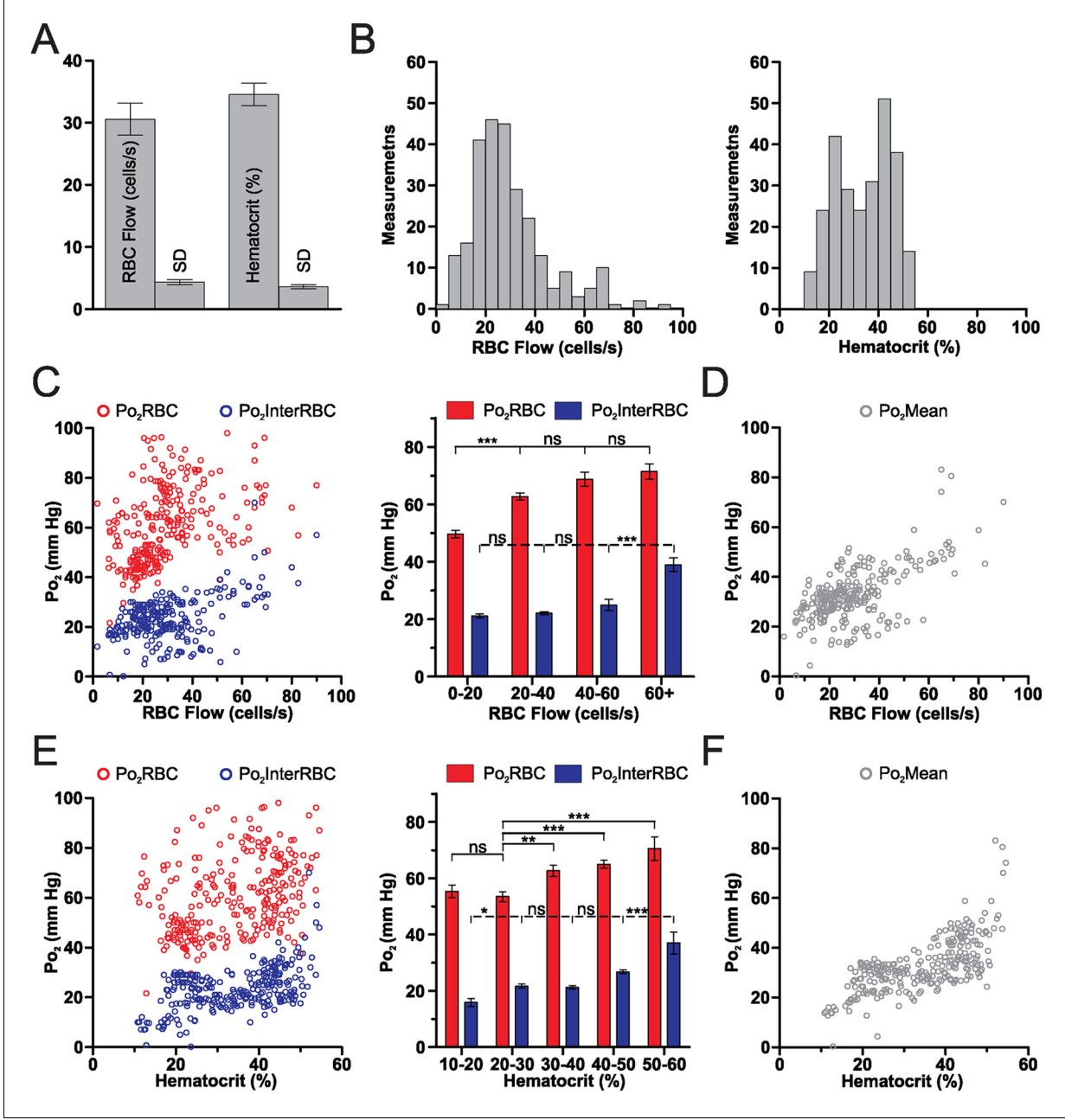

**Figure 2.** Relationships of capillary blood flow and hematocrit to $Po_2$ values, in the olfactory bulb glomerular layer of the awake mouse. (A) RBC flow and hematocrit calculated from the mean values of each capillary (n = 38). SD (the average of the SD values for each capillary, presented as mean standard deviation ± SEM) illustrates the fluctuations of RBC flow and hematocrit in each capillary. (B) Frequency distributions of RBC flow and hematocrit (5% bin). (C) Distribution of all $Po_2RBC$ and $Po_2InterRBC$ measurements as a function of RBC flow. Note that the $Po_2InterRBC$ is independent of RBC flow below 60 cells/s while $Po_2RBC$ increases with RBC flow below 40 cells/s. (D) $Po_2Mean$ as a function of RBC flow. (E) Distribution of $Po_2RBC$ and $Po_2InterRBC$ as a function of hematocrit. $Po_2InterRBC$ is independent of hematocrit from 20 to 50%. $Po_2RBC$ increases with hematocrit at low values. (F) $Po_2Mean$ as a function of hematocrit. Bar graph data presented as mean ± SEM. *p<0.05, **p<0.01, ***p<0.001. Kruskal-Wallis test with 2-tailed Dunn's multiple comparison post-hoc test. For all plots n = 5 mice.

measurements: $Po_2RBC$ = 60.5 ± 0.9 mm Hg; $Po_2InterRBC$ = 23.4 ± 0.5 mm Hg; $Po_2Mean$ = 32.8 ± 0.7 mm Hg. 38 capillaries: $Po_2RBC$ = 60.6 ± 2.3 mm Hg; $Po_2InterRBC$ = 23 ± 1.5 mm Hg; $Po_2Mean$ = 32.7 ± 1.9 mm Hg) (*Figure 1B*). The standard deviation of measurements (SD) made in a given capillary, during single or multiple recording sessions were modest for $Po_2InterRBC$ (mean SD = 2.6 ± 0.3

mm Hg) and slightly larger for $Po_2RBC$ (mean SD = 5.4 ± 0.5 mm Hg). *Figure 1C* illustrates that average $Po_2$ values masked the large span of all values measured. This was particularly true for $Po_2RBC$, for which values frequently exceeded 70 mm Hg. Overall, these data show that in the glomerular layer of the awake resting mouse, interstitial $Po_2$ ranges from 15 to 35 mm Hg in about 82% of our measurements. We then investigated whether the range and fluctuations of $Po_2$ values depend on two vascular parameters, RBC blood flow and hematocrit, in the same capillaries.

## Capillary blood flow and hematocrit in the olfactory bulb of the awake resting mouse

Mean capillary RBC flow and hematocrit values were 30.6 ± 2.6 cells/s and 34.6 ± 1.8%, respectively (*Figure 2A*). Both RBC flow and hematocrit displayed a wide range of values (*Figure 2B*) with a positively skewed frequency distribution of RBC flow. Simultaneous measurements of $Po_2$ and blood flow parameters revealed that although interstitial $Po_2$ ($Po_2InterRBC$) is correlated with both RBC flow and hematocrit (r = 0.3091, p<0.0001 and r = 0.4365, p<0.0001, Spearman r correlation, respectively), these relationships are non-linear. In particular, $Po_2InterRBC$ is mostly independent of both RBC flow below 60 cells/s and hematocrit from 20 to 50% (*Figure 2C and E*), increasing only at high values of both parameters. This stable region covers 82.4% of our measurements. In contrast, $Po_2RBC$ increased with RBC flow and hematocrit at low values, becoming stable above 20 cells/s and 30%, respectively (*Figure 2C and E*). Note that $Po_2Mean$ increased with both RBC flow and hematocrit values (RBC flow: r = 0.5116, p<0.0001; hematocrit: 0.6752, p<0.0001, Spearman r correlation, *Figure 2D and F*).

## Brain oxygenation is greatly enhanced by isoflurane anesthesia

Isoflurane is a volatile anesthetic that is commonly used in the study of brain activation and metabolism. It differently affects regional cerebral blood flow in humans (*Ramani and Wardhan, 2008*), and modulates neurovascular coupling in a concentration-dependent fashion (*Masamoto et al., 2009*) as well as the relationship between spontaneous or evoked neuronal activity with BOLD signal (*Aksenov et al., 2015*). However, its effects on brain oxygenation have only been investigated using approaches with low spatial resolution and which did not allow simultaneous measurement of blood flow (*Liu et al., 1995*; *Ortiz-Prado et al., 2010*). We performed paired measurements of EATs and

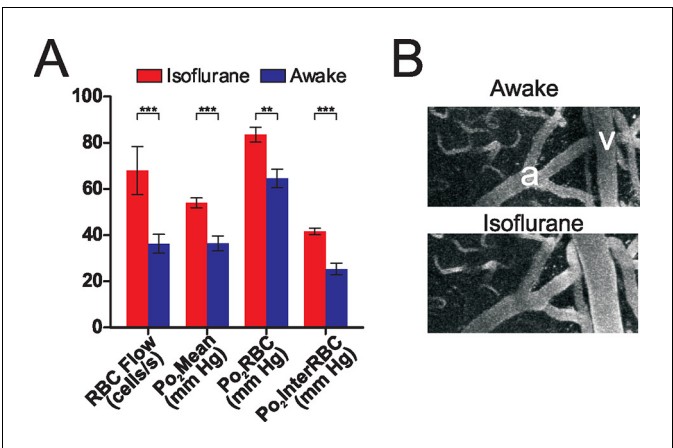

**Figure 3.** Isoflurane changes the brain oxygenation state. $Po_2$ and RBC flow were compared in the same sets of olfactory bulb glomerular layer capillaries when the animal was awake, and when the animal was anesthetized with isoflurane (0.75%, delivered in air, no oxygen added). (**A**) Isoflurane anesthesia increased all RBC flow and $Po_2$ values as compared to the awake state (n = 142 measurements in each condition, 16 capillaries from 3 mice). Bar graph data presented as mean ± SEM., ***p<0.001, paired 2-tailed Wilcoxon signed rank test. (**B**) Isoflurane anesthesia induced a large dilation of pial vessels (a: artery, v: vein). See also *Figure 3—figure supplement 1*.

The following figure supplement is available for figure 3:

**Figure supplement 1.** Isoflurane alters oxygenation in the somatosensory cortex.

flow parameters in a set of capillaries, both when the animals were awake and when they were anesthetized with isoflurane (~0.75% as measured at the animal's snout, delivered in air with no supplementary $O_2$). As can be seen from *Figure 3A*, isoflurane significantly increases all capillary $Po_2$ values ($Po_2Mean$, $Po_2RBC$ and $Po_2InterRBC$). This effect was present in all but one of the capillaries tested. Although reduced neuronal activity (and hence $O_2$ consumption) in the isoflurane anesthetized state (*Aksenov et al., 2015*) is likely to play a role, it appears that this elevation of $Po_2$ largely resulted from an increase in RBC flow, as the increase in $Po_2$ values observed is in accordance with that which would be predicted from the observed increase in RBC flow based on the relationship presented in *Figure 2C and D*. This increase in capillary RBC flow rate is related to isoflurane's vasodilatory effects on large vessels (*Figure 3B*) (*Koenig et al., 1994*). Consequently, cellular processes, including neuronal activity in response to odor, occur at a much higher oxygen concentration during isoflurane anesthesia than in the awake, resting state, and this difference is likely to be exacerbated when isoflurane is delivered in gas mixtures where [$O_2$] is greater than 21% (as in many other studies). Since the olfactory bulb glomerular layer has a specific neuronal and vascular organisation that could be a main determinant of $Po_2$ values, we extended our investigation of tissue oxygenation to the cerebral cortex, due to its importance to higher cognitive functions and its use as an ischemic model.

## Capillary flow and $Po_2$ values in the somatosensory cortex of the awake resting mouse

We made measurements of all $Po_2$ and blood flow parameters in capillaries distributed from the cortical surface to a depth of 410 μm in the fore- and hind-limb regions of the somatosensory cortex. Due to the greater inter-capillary distance in the cortex than in the glomeruli (*Lecoq et al., 2009*; *Sakadžić et al., 2014*), and as $Po_2$ gradients have previously been observed around blood vessels in anesthetized animals (*Devor et al., 2011*; *Sakadžić et al., 2010*; *Sharan et al., 2008*), it is likely that there are tissue regions, in areas far from the nearest capillary, in which the $Po_2$ is lower than what is reported by $Po_2InterRBC$. Nonetheless, based on our previous results (*Parpaleix et al., 2013*) in olfactory bulb glomeruli and on theoretical predictions (*Lücker et al., 2014*), we expect that $Po_2InterRBC$ reports steady-state tissue $Po_2$ up to a radius of approximately 10 μm from a capillary. Furthermore, recent work, that models hematocrit distribution in large microcirculatory networks and accurately replicates physiological RBC distribution, predicts that the distribution of oxygen in tissue volumes supplied by these networks is largely homogeneous (*Gould and Linninger, 2015*). We thus consider that our measured values of $Po_2InterRBC$ represent a significant proportion of the range of $Po_2$ values present in the cortical interstitium. Our $Po_2$ measurements revealed that the averages and ranges of $Po_2$ values in cortical capillaries (*Figure 4A and B*) were similar to those measured in the olfactory bulb glomerular layer ($Po_2RBC$ = 66.3 ± 1.6 mm Hg, $Po_2InterRBC$ = 23.3 ± 1.1 mm Hg, and $Po_2Mean$ = 36.3 ± 1.3 mm Hg), indicating that at rest in the awake state, these two brain areas have similar levels of both RBC, capillary and pericapillary oxygenation. In contrast, the average RBC flow and hematocrit values (41.9 ± 1.8 cells/s and 47.3 ± 0.9%, respectively) were significantly higher than those found in the olfactory bulb glomerular layer (*Figure 4B*) with the hematocrit values being higher than previously reported levels in the cerebral cortex of anesthetized animals (*Hudetz, 1997*). The lower hematocrit levels in glomerular layer capillaries could be related to differences in the bulb cerebrovascular supply which, in contrast to the cortex (*Blinder et al., 2013*), is poorly established (*Coyle, 1975*).

As in the olfactory bulb, both $Po_2RBC$ and $Po_2InterRBC$ were correlated with RBC flow rate and hematocrit ($Po_2RBC$ with RBC Flow and hematocrit: r = 0.3824, p<0.0001 and r = 0.2315, p<0.0001, Spearman r correlation, respectively; $Po_2InterRBC$ with RBC Flow and hematocrit: r = 0.4283, p<0.0001 and r = 0.5018, p<0.0001, Spearman r correlation, respectively). $Po_2InterRBC$ (*Figure 4D and F*) increased with RBC flow, from 20 to 60 cells/s, that is, over the majority of the measurements, becoming stable thereafter. It also increased with hematocrit. Similarly, $Po_2RBC$ increased with both RBC flow and hematocrit. Note that $Po_2Mean$, was correlated with and increased through the entire range with both RBC flow and hematocrit (RBC flow: r = 0.4670, p<0.0001; hematocrit: 0.5945, p<0.0001, Spearman r correlation, *Figure 4E and G*).

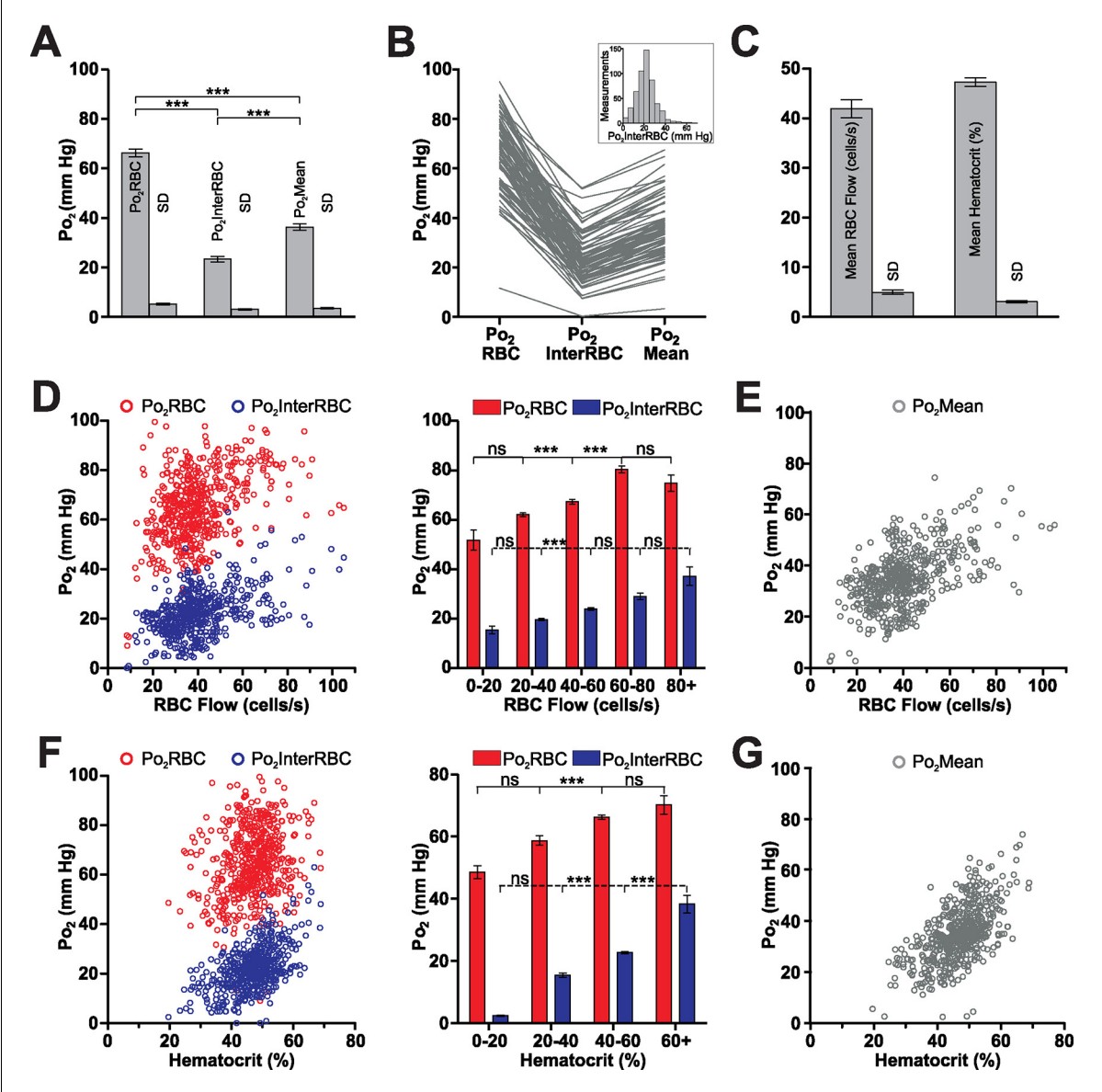

**Figure 4.** The relationship of $Po_2$ to RBC flow and hematocrit in the somatosensory cortex of the awake mouse. (**A**) Average values and SD of $Po_2$ parameters in somatosensory cortex capillaries, calculated from the mean values from each capillary (81 capillaries, 528 measurements, SD = the average of the SD values for each capillary, presented as mean standard deviation ± SEM). (**B**) Distribution of all capillary $Po_2$ values averaged in (**A**). Frequency distribution histogram of local tissue $Po_2$ ($Po_2$InterRBC) in inset. 5 mm Hg bin. (**C**) Average values and SD of RBC flow and hematocrit calculated from the mean values from each capillary. (**D**) Distribution of all $Po_2$RBC and $Po_2$InterRBC measurements as a function of RBC flow. Note that for most values (from 20 to 60 cells/s), both $Po_2$InterRBC and $Po_2$RBC increase with RBC flow. (**E**) $Po_2$Mean as a function of RBC flow. (**F**) Distribution of all $Po_2$RBC and $Po_2$InterRBC measurements as a function of hematocrit. For most values (from 20 to 60% ), both $Po_2$InterRBC and $Po_2$RBC increase with hematocrit. (**G**) $Po_2$Mean as a function of hematocrit. Bar graph data presented as Mean ± SEM. *$p<0.05$, **$p<0.01$, ***$p<0.001$. Kruskal-Wallis test with 2-tailed Dunn's multiple comparison post-hoc test.

## Laminar organisation of capillary blood flow and Po₂ values in the superficial layers of the somatosensory cortex

Several studies, using polarographic electrodes or 2PLM in anesthetized rodents, have reported that vascular and interstitial $Po_2$ varies with cortical depth (*Devor et al., 2011*; *Masamoto et al., 2003*; *Sakadžić et al., 2010*). As anesthetics could differently modulate synaptic activity and blood flow in layers I to IV, it is difficult to predict the extent to which $Po_2$ depth variations occur in the awake

animal. We thus first measured $Po_2$ in descending and ascending large vessels and then compared $Po_2$ and blood flow parameters of capillaries from layers I to IV (from the surface to 410 μm in depth). Capillaries less than 60 μm below the surface were considered to be in layer I, those from 90 to 260 μm below the surface were classified as layer II/III capillaries and those deeper than 340 μm were considered layer IV capillaries (capillaries from 260 to 340 μm were not considered due the ambiguity of their location). Penetrating vessels (arterioles and venules) were traced from their point of descent below the surface down along their main trunk until they ramified into smaller vessels (or descended below 410 μm, which was the maximum depth at which we made measurements). Note that in these vessels EATs were not detectable due to the close apposition of RBCs. The mean $Po_2$ was 69.2 ± 1.4 mm Hg for arterioles and 39.9 ± 1.3 mm Hg for venules and, in contrast to what was reported in anesthetized animals (*Devor et al., 2011*; *Sakadžić et al., 2010*), no gradient was observed with depth (*Figure 5A and B*).

Laminar analysis of capillary blood flow and $Po_2$ values revealed some specific differences: although the hematocrit and blood flow values were similar in all three layers (*Figure 5C*, right panel), all $Po_2$ values were lower in layer I than in layer II/III (*Figure 5C*, left panel). The low $Po_2$InterRBC values 14.7 ± 1.7 mm Hg suggests that interstitial $Po_2$ is correlated with the capillary density which is lower in layer I than II/III (*Blinder et al., 2013*; *Sakadžić et al., 2014*). The correlated analysis of $Po_2$ with hematocrit revealed that low $Po_2$InterRBC values were present at a wide distribution of hematocrit levels (*Figure 5D*, left panel). In addition, even though the average $Po_2$RBC was higher in layer II/III than in layer IV, it was independent of hematocrit in both layers (*Figure 5D*). Finally, the effects of isoflurane in the cortex (Layer I, *Figure 3—figure supplement 1*) were similar to those observed in the olfactory bulb. Thus, in addition to its direct effects on neurons, isoflurane increases oxygen delivery to the entire brain.

## Regions of low tissue $Po_2$ are present at rest

$Po_2$InterRBC values show a wide distribution in both the olfactory bulb glomerular layer and the somatosensory cortex (*Figure 1C* and *4B*). Notable in both structures is the presence of measurements of $Po_2$InterRBC, and hence local tissue $Po_2$, that were <15 mm Hg. In the olfactory bulb glomerular layer these low $Po_2$InterRBC capillaries (n=5) were found to have a large range of RBC flow rates but low hematocrit levels (<25%, *Figure 6A and 6B*). In contrast, in somatosensory cortex capillaries (n = 13) in which the $Po_2$InterRBC value was <15 mm Hg, neither RBC flow rates nor hematocrit levels were notably low (*Figure 6C and 6D*). Instead, these low $Po_2$InterRBC values were mostly found in capillaries in layer I (10 of 13 capillaries), suggesting that the presence of low tissue $Po_2$ values results from several factors.

## RBC hemoglobin is largely saturated in cerebral capillaries

Given that $Po_2$RBC should be representative of the $Po_2$ level inside RBCs, we used known values of the Hill coefficient and $P_{50}$ for mouse hemoglobin (*Uchida et al., 1998*) to estimate hemoglobin saturation ($So_2$) in capillaries. In both the olfactory bulb glomerular layer and the somatosensory cortex (*Figure 7A and B*), measured $So_2$ is >50% in the vast majority of cases (94.7% and 98.5% of measurements in the glomerular layer and the cortex, respectively). This shows that in the resting brain, the majority of hemoglobin exists as oxyhemoglobin. Furthermore the presence of numerous measurements of $So_2$ >85% in both structures suggests that hemoglobin saturation is nearly maximal in a significant proportion of capillaries (~10% of capillaries in the glomerular layer, ~20% in the somatosensory cortex). Note that, if $So_2$ were to be inappropriately estimated from $Po_2$Mean values, a very different distribution of hemoglobin saturation would be derived (*Figure 7C*). This demonstrates that in capillaries, an accurate measurement of $Po_2$RBC is a prerequisite for estimating $So_2$.

## Discussion

In the olfactory bulb and somatosensory cortex of awake resting mice, we report similar average values of interstitial $Po_2$ (~23 mm Hg). In the awake rodent, brain oxygenation has previously only been investigated with lower resolution approaches. Liu et al. (*Liu et al., 1995*) used electron paramagnetic resonance (EPR) oximetry with a lithium phthalocyanine crystal being implanted in the cerebral cortex of rats 24 hr prior to the measurements. They reported that restrained and untrained rats have a cerebral $Po_2$ of about 34 mm Hg, which, surprisingly, was reduced by isoflurane anesthesia

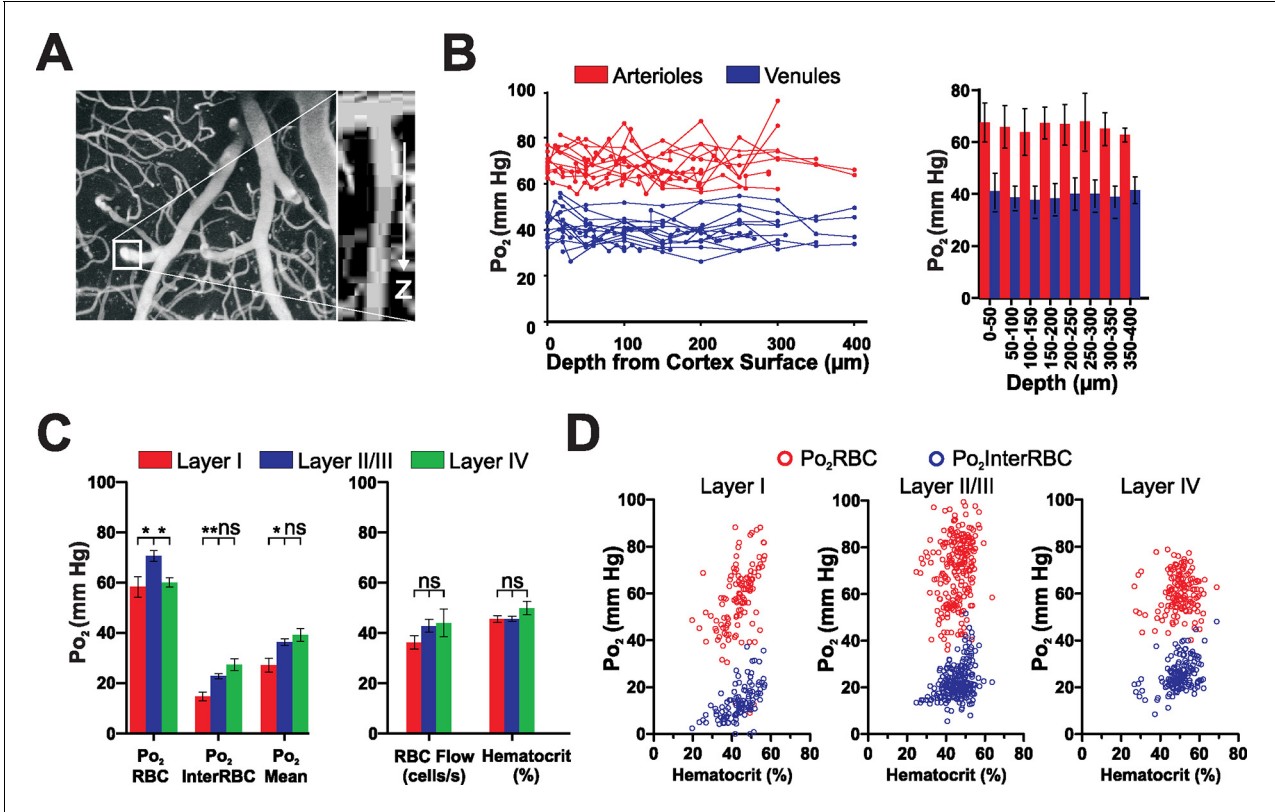

**Figure 5.** Depth profiles of vascular and local tissue oxygenation in the somatosensory cortex of the awake mouse. (**A**) Left panel, maximum intensity projection of superficial portion of the vasculature of the somatosensory cortex, with boxed area highlighting a penetrating arteriole shown in the XZ projection (right panel). (**B**) Left panel, $Po_2$ values in penetrating arterioles (red) and venules (blue) as a function of depth from the cortical surface. Each line represents a single vessel. Right panel, mean of all $Po_2$ values recorded from vessels as a function of depth. Note the absence of $Po_2$ gradients with depth (50 μm bin size, 136 measurements in 11 arterioles, 148 measurements in 14 venules, from 6 mice.) Data presented as mean ± SD (**C**) Comparison of $Po_2$ values (left panel), RBC flow and hematocrit (right panel) in layers I, II/III and IV. Note that capillary $Po_2Mean$ and $Po_2InterRBC$ are higher in layers II/III and IV than in layer I, although there are no significant differences in either blood flow parameter. Data presented as mean ± SEM. * $p<0.05$, **$p<0.01$, Kruskal-Wallis test with 2-tailed Dunn's multiple comparison post-hoc test. n = Layer I: 113 measurements in 17 capillaries, Layer II/III: 230 measurements in 41 capillaries, Layer IV: 151 measurements in 15 capillaries (**D**) $Po_2RBC$ and $Po_2InterRBC$ as a function of hematocrit in layers I, II/III and IV.

(1% in 21% $O_2$) to about 24 mm Hg. This resting value (34 mm Hg) is significantly higher than our average interstitial $Po_2$ value, most probably due to the effects of the stress resulting from restraint in the absence of habituation. The effect of isoflurane, which decreased $Po_2$ is intriguing, and indicates that the main effect of isoflurane in this case was to abolish the $Po_2$ augmenting effects of stress. Interestingly, increasing the delay from crystal implantation to 7 days and introducing some habituation to restraint reduced cerebral $Po_2$ to 27 mm Hg (*Dunn et al., 2000*). The best demonstration of physiological measurements of $Po_2$ has been performed in unrestrained rats, in which brain $Po_2$ was measured with an implanted fiber optic probe measuring quenching of an oxygen sensor (*Ortiz-Prado et al., 2010*). Although the probe was very invasive, it reported an average bulk tissue $Po_2$ value of 25 mm Hg, a value very similar to ours. We are thus confident that $Po_2InterRBC$ is an excellent reporter of the interstitial $Po_2$, that our extensive habituation procedures are efficient, and that this study gives the first non-invasive and physiological values of $Po_2$, hematocrit and capillary blood flow in the rodent brain.

Our results show a number of differences from those previously reported with high-resolution measurement techniques in anesthetized animals. Tissue $Po_2$ values have previously been reported to be in the range of ~5–100 mm Hg, with the higher values occurring in regions close to pial arterioles (*Devor et al., 2011*; *Sakadzić et al., 2010*). Our values of tissue $Po_2$ ($Po_2InterRBC$) lie in the lower end of this wide range, do not include measurements from the capillary-free peri-arteriolar

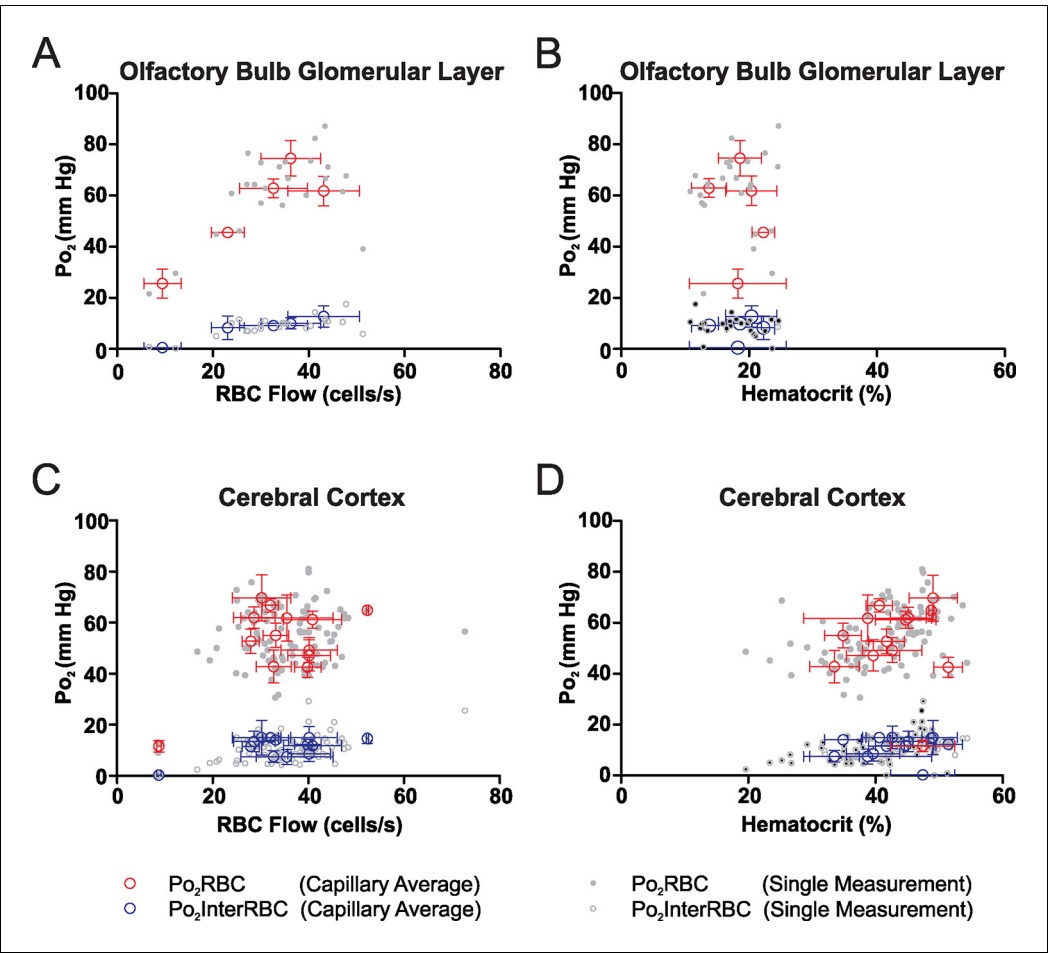

**Figure 6.** In the awake mouse, low interstitial Po$_2$ in the olfactory bulb glomerular layer is associated with low hematocrit capillaries. Po$_2$ values from all glomerular layer capillaries with average Po$_2$InterRBC values of <15 mm Hg (n = 24 measurements in capillaries) are plotted as a function of RBC flow (**A**) and hematocrit (**B**). The RBC flow rates in these capillaries were distributed across a wide range, whereas in all cases capillary average hematocrit was <25%, suggesting that, in the glomerular layer, areas of low interstitial Po$_2$ are supplied by capillaries with relatively low hematocrit values (hematocrit of these capillaries = 18.6 ± 1.4%, n = 5; hematocrit of all other capillaries = 37 ± 1.7%, n = 33; p = 0.002, unpaired t-test. mean ± SEM). Conversely, cortical capillaries with average Po$_2$InterRBC <15 mm Hg (n = 13 capillaries, 101 measurements) had wide ranges of both RBC flow (**C**) and hematocrit (**D**). However, the majority (10 of 13) of these capillaries were located in layer I. In all plots, single measurement values and mean ± SD of all measurements in each capillary shown.

regions, and so are likely to be reflective of the Po$_2$ levels that exist in the majority of the tissue, away from large arterioles. Considering intravascular Po$_2$, our measurements of capillary Po$_2$Mean (average ≈36 mm Hg in the cortex) seem to be broadly in agreement with those reported by *Sakadžić et al., 2010* (~25–35 mm Hg), but our recorded values of capillary RBC Po$_2$ (Po$_2$RBC, average ≈66 mm Hg in the cortex), are dramatically higher than those published by *Sakadžić et al., 2014*, where the most frequent values were ~25–35 mm Hg. A prominent discrepancy between the results of previous studies and our findings in awake, resting animals relates to changes in cortical oxygenation with depth. Many previous studies report drops in interstitial Po$_2$ with increasing depth in the cortex (*Cross and Silver, 1962*; *Devor et al., 2011*; *Masamoto et al., 2003*; *Nair et al., 1975*; *Ndubuizu and LaManna, 2007*; *Whalen et al., 1970*), with Po$_2$ in layer I being higher than in underlying layers. In contrast, our measures of Po$_2$InterRBC suggest that interstitial Po$_2$ is lower in layer I than in either layer II/III or IV. These conflicting patterns of laminar variations of Po$_2$ also exist in comparisons of the mean vascular Po$_2$ values (Po$_2$Mean), where previous 2PLM studies have reported decreases in mean vascular Po$_2$ in concert with decreases in penetrating arteriole and

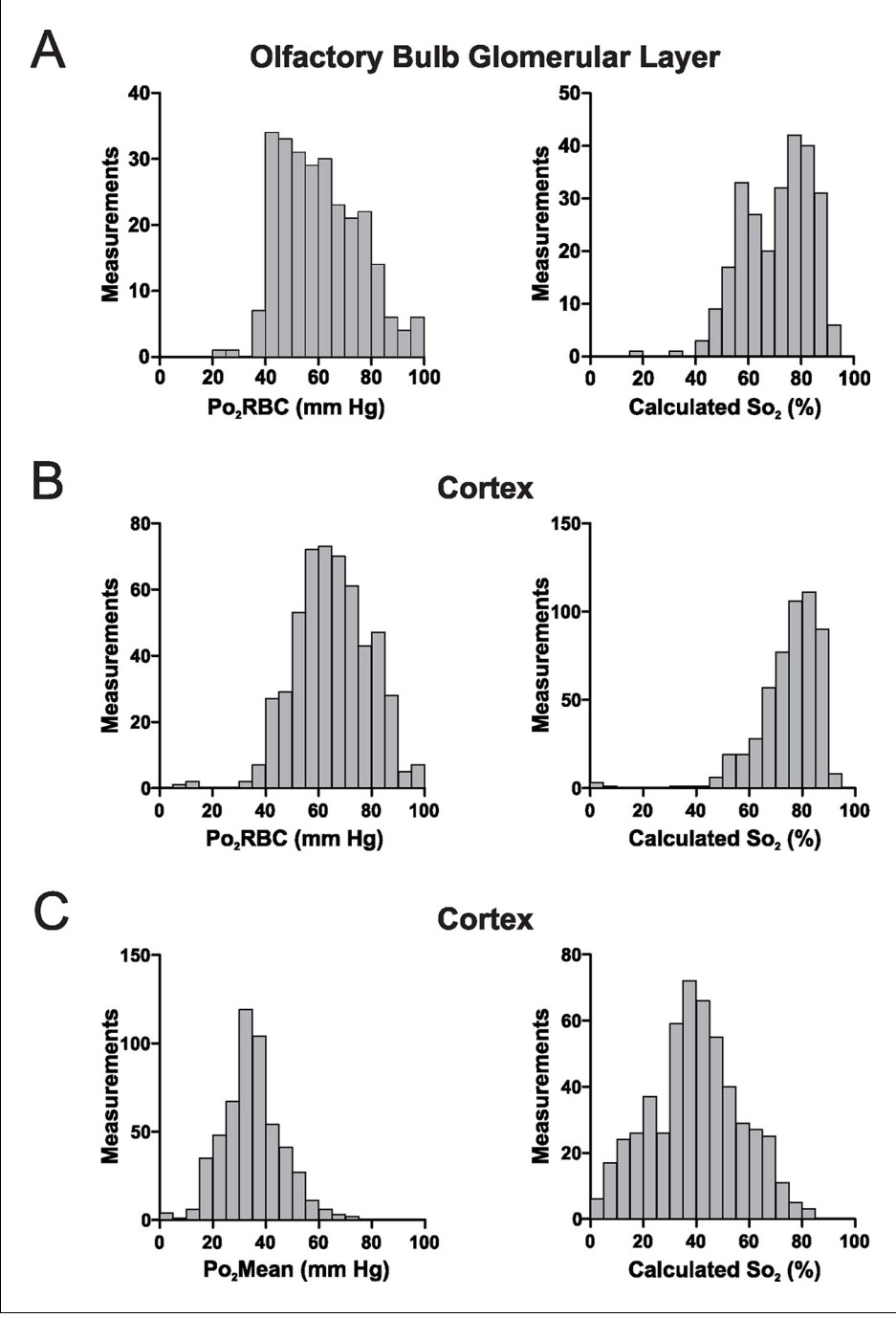

**Figure 7.** In the awake mouse, the majority of hemoglobin in cerebral capillaries is oxygenated. (**A**) Left panel, frequency distribution of $Po_2RBC$ values measured in the glomerular layer of awake mice, which were used to compute $So_2$ values (right panel). (**B**) Equivalent $Po_2RBC$ (left panel) and $So_2$ distributions (right panel) from the cerebral cortex. In both structures > 90% of the measurements have $So_2$ values of >50% (94.7% and 98.5% of measurements in the glomerular layer and the cortex respectively) with ~60% of $So_2$ values in the cortex being >75%. (**C**) Left panel, frequency distribution of $Po_2Mean$ values from cerebral cortex capillaries, with the $So_2$ distribution that would be computed were these lower $Po_2$ values considered to represent those present at the hemoglobin molecules in RBCs (right panel). Glomerular layer: n = 262 measurements in 38 capillaries. Somatosensory cortex: n = 528 measurements from 81 capillaries. Bin size 5 mm Hg and 5% for $Po_2$ and $So_2$ respectively.

venule $Po_2$ (*Devor et al., 2011*; *Sakadzić et al., 2010*) with increasing depth in the cortex. In the present study, we see an increase in capillary $Po_2$Mean from layer I to deeper layers, and no gradient in penetrating vessel $Po_2$ with depth. It is possible that the deviation from normal physiological conditions inherent in anesthesia and acute surgical preparation, which we avoid with our approach, leads to the emergence of these disagreements.

Although average $Po_2$InterRBC values in both the cortex and glomerular layer were ~23 mm Hg, in both structures a number of capillaries were found to have $Po_2$InterRBC values <15 mm Hg, indicating the existence of regions in both structures where the interstitial $Po_2$ is close to reported values of $Po_2$ below which cellular respiratory rate is strongly dependent on $Po_2$ (~10 mm Hg) (*Kasischke et al., 2011*). Furthermore, examples were found in both brain regions where the $Po_2$InterRBC was below the value of ~3.4 mm Hg that has previously been reported as the critical $Po_2$ in brain tissue (*Kasischke et al., 2011*). Similarly low tissue $Po_2$ measurements have previously been observed (*Ndubuizu and LaManna, 2007*; *Whalen et al., 1973*), but their existence has been interpreted as being related to disruption of normal tissue physiology in the experimental preparations (*Wilson et al., 2012*). However, the presence of such low values in our awake preparations indicates that, surprisingly, at least some small regions of the brain can subsist at very low $Po_2$ values.

Our finding of high $Po_2$RBC and corresponding capillary $So_2$ (estimated using the parameters used in *Sakadzić et al., 2014*) values in both the olfactory bulb and the somatosensory cortex (*Figure 7*) differs from recently published findings in the anesthetized, ventilated mouse (*Sakadzić et al., 2014*). We find that capillary $So_2$ in both structures is generally high, indicating that the majority of hemoglobin in these vessels exists as oxyhemoglobin, and thus that capillaries are capable of supplying very significant quantities of oxygen to support neural function.

In conclusion, the present study establishes, for the first time, accurate and precise values of physiological $Po_2$ in the vasculature and interstitium of mouse cerebral grey matter. As it is known that $O_2$ concentration is a critical parameter in determining the properties of neuronal function (*Huchzermeyer et al., 2008*, *2013*; *Ivanov and Zilberter, 2011*), and neurovascular interactions (*Gordon et al., 2008*), these values provide a standard on which future research can rely to provide relevant, physiologically accurate conditions of oxygenation in which to investigate such processes.

## Materials and methods

### Experimental procedures

#### Animal preparation and surgery

All animal care and experimentation was performed in accordance with the INSERM Animal Care and Use Committee guidelines. Adult C57BL/6 mice (3–6 months old, 25-–35 g, both males and female, housed in 12-hr light-dark cycle) were used in this study (n = 5 mice for olfactory bulb capillaries, n = 3 mice for cortex capillaries). Chronic glass cranial windows were implanted over the area of interest, either the olfactory bulb or the somatosensory cortex, with great care taken to not disturb the dura mater. The animals received anti-inflammatory and analgesic treatment (Carprofen, one daily 0.15 mg subcutaneous injection, administered pre-surgically and for the three days post-surgery), antibiotics (Ceftrioxone, one daily 0.25 mg subcutaneous injection, administered pre-surgically and for the three days post-surgery) and dexamethasone to reduce cerebral edema (one daily 60 μg subcutaneous injection on the day before surgery, directly before the surgery and the first post-surgical day). Surgical anesthesia was induced with ketamine-xylazine (100 mg and 8 mg per kg body mass, respectively). During surgery, the mice breathed a mixture of air and supplementary oxygen (the final inhaled proportion of oxygen was ~30%) and the body temperature monitored by a rectal probe and maintained at ~36.5°C by a feedback-controlled heating pad. A craniotomy was performed with great care taken not to apply great pressure to the bone and the area was regularly flushed with cool aqueous buffer solution to avoid damage or heating of the underlying tissue. Cover glass (~150 μm thick) was used for the window and sealed in place with photopolymerizable dental cement (Tetric EvoFlow, Ivoclar Vivadent, Schaan, Principality of Liechtenstein), which was also used to form a head-cap in which a titanium head-bar was also embedded.

## Habituation of mice to head restraint

In the weeks preceding surgery, the animals were supplied with a treadmill (Fast-Trac, Bio-Serv, Flemington, New Jersey) in their cages that was similar to that which forms part of the restraint apparatus used during 2PLM recording (see below). In the days that preceded the surgery, the mice were gently habituated to handling, and provided with treats (sugar pellets, Bio-Serv). 2–3 days after the surgery restraint habituation began. The frame used for head restraint during the habituation and imaging consists of a metal frame in which the mouse's head-bar is secured, and a treadmill wheel (similar to that in the animal's home cage). This configuration allows the animal limbs and body to move freely while the head is stably fixed. All restraint-habituation sessions were carried out in the set-up that was used for imaging, with the animal kept in the near-dark at all times. Habituation sessions were performed multiple times per day over the course of 2–4 weeks, with the duration increasing from 5 min to >1 hr. Animals were considered ready for use in experiments when they could be easily fixed in the recording apparatus while awake, and their behavior during the sessions consisted of short bouts of locomotion (~30 s) separated by longer periods of stillness (5–15 min) during which measurements were performed.

On the days on which recordings were taken, the animals were briefly anesthetized with isoflurane (2%, in air. Total duration of anesthesia <5 min), and PtP-C343 solution (Mw≈65 kDa, 2.5 mM in 0.9% saline) was introduced intravenously via retro-orbital injection. Fluorescein isothiocyanate dextran (Mw = 150 kDa) was often co-administered with PtP-C343 to enhance imaging contrast. The final PtP-C343 concentration in the plasma is estimated to be ~100 μM. After the injection, the mice were returned to their home cages for ~90 min, before being brought back to the experimental room, for ~30 min before the start of the recording session. We implemented this delay of ~2 hr between the injection and the following recording session to avoid potential confounding effects of the brief exposure to isoflurane or the effects of the injection itself. We found that the concentration of PtP-C343 in the blood typically remained sufficient for $Po_2$ measurements to be made over 1–3 days after an injection, allowing for multiple recording sessions to be made before reinjection was necessary. Each recording session lasted from 1–3 hr, depending on the type of measurements performed and the behavior of the animal, and each animal underwent 1–3 recording sessions per day over the course of 2–7 days, with breaks of at least 1–3 hr between each session. The data derived from each of the recording sessions across the 1–2 days post-injection was similar, indicating that no lingering effects of the injection procedure or brief isoflurane anesthesia had affected the measurements in the first recording session post-injection.

## 2PLM setup and procedure

The 2PLM setup is as described in *Lecoq et al. (2011)*, and *Parpaleix et al. (2013)*. Briefly, the output from a Ti:sapphire laser (Mira, Coherent, Santa Clara, California; λ = 850 nm, 120 fs pulse width, 76 mHz) is gated by an acousto-optic modulator (AOM, AA Optoelectronic, Orsay, France; MT110-B50-A1.5-IR-Hk). This allows for repetitive cycles of alternating 'on' and 'off' periods to be generated, that correspond to periods of excitation and recording from the PtP-C343 probe in the sample. The excitation period was 24 μs and the recording period was 225 μs, for a total cycle period of 250 μs and a repetition rate of 4 kHz. The scanning of the excitation light by the galvanometric mirrors (Cambridge Technology, Bedford, Massachusetts) is synchronized with the gating of the laser output by the AOM, using custom-built electronics and LabVIEW software (National Instruments, Austin, Texas). The excitation light was focused with a water-immersion objective lens (either x63 Leica [Wetzlar, Germany] or x40 Olympus lens [Olympus, Tokyo, Japan]).

Emitted photons were divided by a dichroic mirror (cut off wavelength = 560 nm). The green channel was bandpass filtered (HQ 520/40 m, Chroma Technology Corp, Bellows Falls, Vermont). The red channel had two shortpass filters (FF01-750/sp-25, Semrock, Rochester, New York), and one bandpass filter (HQ 680/60 m-2P, Chroma Technologies Corp), and a red-sensitive photomultiplier tube (PMT, R10699, Hamamatsu, Hamamatsu City, Japan). PMT signals were amplified with custom-build electronics and sampled at 1.25 MHz by an acquisition card. Photons detected in the green channel during the on phase of the recording cycle were used to detect the borders of RBCs and extract RBC flow rates and hematocrit values (see below). Phosphorescence decays detected during the 'off' period were averaged over a number of cycles (~5000–50,000 decays) and the lifetime of the phosphorescence determined by fitting a single exponential curve to the data. The first 6–7 bins

(~5 μs) after the end of the on-phase were discarded. This lifetime measurement is then converted to a value of $Po_2$ using a calibration curve. We used the nonlinear least-squares method and the Marquardt-Levenberg algorithm to obtain the decay lifetimes and associated standard errors. We used a Stern-Volmer–like calibration curve to convert the phosphorescence lifetimes into the corresponding $Po_2$ values. This analysis was performed using custom build software (see below) developed using LabView (National Instruments).

## Capillary blood flow analysis and EAT properties extraction

Initially, fast 3D image acquisition was performed to identify and target vessels of interest. $Po_2$ measurements were then made by focusing the excitation point in blood vessels of interest, and recording the fluorescence and phosphorescence emitted from 10,000–40,000 cycles of excitation and detection of phosphorescence decays (2.5–10 s of measurement) at each point. Typically 4–8 measurements were performed in each capillary, with the total number of decays recorded for each capillary being around 60,000–100,000.

Details of the methods of determining EAT properties, RBC flow rates and hematocrit can be found in *Parpaleix et al. (2013)* and *Lecoq et al. (2011)*, see especially *Figures 1* and *3* therein). In brief, the fluorescence recorded during the excitation period of the cycle was analyzed and the passage of RBCs through the excitation point detected based on the associated dips in fluorescence intensity. These transient reductions in fluorescence intensity were detected and quantified using a binary threshold method, with the borders of the RBC corresponding to the full width at half maximum of the fluorescence dip. This allowed for the measurement of both capillary RBC flow rate (the number of RBCs detected per unit time) and hematocrit (the number of decay cycles originating from within RBCs as a percentage of the total number of decay cycles). The borders of the detected RBCs were used as time-stamps, and the phosphorescence decays recorded were binned according to their distance in time from the borders of the nearest RBC. The $Po_2RBC$ value was calculated from the average lifetime of decays recorded in the 1–4 ms around the border of the RBC, whereas the $Po_2InterRBC$ was determined from average value of decays at mid-distance between RBCs (averaged over a window of at least 5 ms). The EAT-extraction custom-built software is now available online at the following address: https://github.com/charpak-lab/EAT-detection. Hemoglobin saturation ($So_2$) was estimated from $Po_2RBC$ using the Hill equation, with a Hill coefficient (2.59) and $P_{50}$ (40.2 mm Hg), which are accurate for C57BL/6 mice (*Uchida et al., 1998*) and have previously been used to make estimations of mouse cerebral $So_2$ from $Po_2$ data (*Sakadžić et al., 2014*).

## Acknowledgements

We would like to particularly thank Sergey Vinogradov for providing PtP-C343. We also thank Emmanuelle Chaigneau, Mathieu Ducros, Yannick Goulam-Houssen and Sergei Sasnouski for software support, along with David Attwell, Ravi Rungta and Etienne Audinat for their critical comments.

## Additional information

### Funding

| Funder | Grant reference number | Author |
|---|---|---|
| Institut National de la Santé et de la Recherche Médicale | | Serge Charpak |
| Fondation Leducq | | Serge Charpak |
| European Research Council | ERC-2013-AD6 (339513) | Serge Charpak |
| Ecole de Neuroscience, Paris Ile-de-France | | Serge Charpak |
| Fondation pour la Recherche Médicale | Equipe FRM | Serge Charpak |
| Ecole de Neuroscience, Paris Ile-de-France | Doctoral Fellowship | Declan G Lyons |

| Fondation pour la Recherche Médicale | FDT20130928252 | Declan G Lyons |
| --- | --- | --- |
| Agence Nationale de la Recherche | ANR 12 BSU4-0017-01 | Serge Charpak |

The funders had no role in study design, data collection and interpretation, or the decision to submit the work for publication.

## Author contributions

DGL, Conducted the experiments, Conception and design, Acquisition of data, Analysis and interpretation of data, Drafting or revising the article; AP, Conducted the experiments, Conception and design, Acquisition of data; MR, Conducted the experiments, Acquisition of data; SC, Conducted the experiments, Conception and design, Analysis and interpretation of data, Drafting or revising the article

## Author ORCIDs

Declan G Lyons, http://orcid.org/0000-0003-1775-4653

## Ethics

Animal experimentation: All animal care and experimentation was performed in accordance with the INSERM Animal Care and Use Committee guidelines (protocole number CEEA34.SC.122.12 and CEEA34.SC.123.12)

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
