## [Decision Letter]

Thank you for submitting your work entitled "Mapping oxygen concentration in the awake mouse brain" for consideration by *eLife*. Your article has been reviewed by three peer reviewers, and the evaluation has been overseen by David Kleinfeld as the Reviewing Editor and Timothy Behrens as the Senior Editor.

The reviewers have discussed the reviews with one another and the Reviewing editor has drafted this decision to help you prepare a revised submission. All reviewers are enthusiastic about this work. Each raised a number of questions. I ask you address each question by clarifying the manuscript. No new data is required. I have summarized the key questions:

1) The hematocrit is reported to be significantly higher in cortex than in olfactory bulb (subsection “Capillary flow and Po_2_ values in the somatosensory cortex of the awake resting mouse”). This needs to be discussed – how is this difference produced? Is it somehow produced by plasma skimming at a series of junctions of the vessels supplying the olfactory bulb? As this is an acute preparation, one needs to be certain that this is not due to some small amount of tissue damage when making the imaging window.

2) Please show a raw trace of the Po_2_ profile between RBCs, or state explicitly that the line in the inset to panel A IS raw data (and not a schematic); why is that line continuous to the left and dotted on the right hand side?

3) The authors repeatedly stress that previous measurements using two-photon phosphorescence only report Mean Po_2_. Could the authors please include a comparison of their Mean Po_2_ values against those made by other groups also using two-photon phosphorescence probe?

4) Please explain the bars in Figure 1 far better. State completely clearly that Po_2_ is the mean (over vessels) of a single time averaged value obtained from each vessel, while SD is mean (over vessels) of the standard deviation of the Po_2_ in each vessel over time (if that is what it is!).

5) The Po_2_ seems (subsection “Laminar organisation of capillary blood flow and Po_2_ values in the superficial layers of the somatosensory cortex”) to be determined by the local capillary density. Does this imply that local O_2_ consumption is independent of laminar position (one might have expected vessel density to be proportional to local O_2_ usage)? Is it possible to derive, from the laminar distribution of [O_2_] and the blood flow reaching each lamina, how much O_2_ is consumed by each cell layer?

6) A more thorough clarification of the argument of how isoflurane increases interstitial oxygen concentration would strengthen the manuscript. The current explanation links increased bloodstream oxygenation to the vasodilatory effect of isoflurane (subsection “Brain oxygenation is greatly enhanced by isoflurane anesthesia”). However, the increase in RBC flux (Figure 3) resulting from vasodilation would lead to a decrease in oxygen unloading from RBCs, as evidenced by the increase in RBC Po_2_ (Figure 3). If less oxygen is unloaded from the red blood cells, then less oxygen diffuses through the vessel wall, leading to a decrease in tissue oxygenation. This is contradicted by the statement that cellular processes occur at a higher oxygen concentration during isoflurane anaesthesia. This matter is further confused by the delivery of isoflurane in a 21% oxygen gas mixture.

7) Theoretical models are cited as evidence that InterRBC Po_2_ is in equilibrium with the surrounding tissue (subsection “Capillary flow and Po_2_ values in the somatosensory cortex of the awake resting mouse”). The theoretical models rely on accurate hematocrit prediction to determine the distribution of oxygen to the surrounding tissue. The relationship between radial oxygen gradients around capillaries, hematocrit and RBC velocities have been discussed in (I. Gould, A. Linninger, "Hematocrit Distribution and Tissue Oxygenation in Large Microcirculatory Networks", Microcirculation, 22, p1-18, 2015). The authors may consider inclusion of this reference to support the argument.

8) The authors mention that they anaesthetise all mice before initiating even awake imaging experiments. Is it possible to head fix the mice without using anaesthesia? Is the anaesthesia necessary to do the retro-orbital injection of the fluorescein dextran and oxygen probe? It seems awake mice could be tail/vein injected. This might also affect the results since there could residual effects of anaesthesia. At the very least this confound should be acknowledged.

The reviewers have also identified some technical and typographic errors that you should attend to. See the full reviews attached below. We look forward to a revised submission together with a tally of brief answers to each of the seven points above. This is beautiful work and we hope to see it published soon.

*Reviewer #1:*

This manuscript provides important data (obtained using a relatively non-invasive technique) characterising the O_2_ level at different parts of the vasculature in awake non-anesthetized animals. Although mainly descriptive, the data will be of great interest to all those working on the control of cerebral blood flow, energy supply to the brain, brain metabolism and BOLD fMRI signals, because all of these subjects depend critically on the local O_2_ concentration, which is poorly defined in vivo. Previous work has only provided similar data on anesthetized animals (or with more invasive techniques in freely moving animals), and it turns out, importantly, that the values are very different in the unanesthetized animal. This paper is therefore a significant step forward, and I support its publication.

There are some areas where the discussion should be improved, as follows.

1) The hematocrit is reported to be significantly higher in cortex than in olfactory bulb (subsection “Capillary flow and Po_2_ values in the somatosensory cortex of the awake resting mouse”). This needs to be discussed – how is this difference produced? Is it somehow produced by plasma skimming at a series of junctions of the vessels supplying the olfactory bulb? Why might that occur more for the supply to the olfactory bulb than for that to the cortex?

2) The Po_2_ seems (subsection “Laminar organisation of capillary blood flow and Po_2_ values in the superficial layers of the somatosensory cortex”) to be determined by the local capillary density. Does this imply that local O_2_ consumption is independent of laminar position (one might have expected vessel density to be proportional to local O_2_ usage)? Is it possible to derive, from the laminar distribution of [O_2_] and the blood flow reaching each lamina, how much O_2_ is consumed by each cell layer?

3) It is surprising that this study finds a lower Po_2_ in the surface layers of the cortex, whereas Devor and Sakadzic found a decrease in [O_2_] with depth. Is there any methodological difference that could account for this, e.g. O_2_ penetration into the brain from the cortical window in the earlier studies)? Was the laminar profile of O2 affected by isoflurane?

4) Figure 1. Please show a raw trace of the Po_2_ profile between RBCs, or state explicitly that the line in the inset to panel A IS raw data (and not a schematic); why is that line continuous to the left and dotted on the right hand side?

5) Figure 1. Please explain the bars far better. State completely clearly that Po_2_ is the mean (over vessels) of a single time averaged value obtained from each vessel, while SD is mean (over vessels) of the standard deviation of the Po_2_ in each vessel over time (if that is what it is!). Please try out your text on a naive reader so that it is comprehensible.

6) Figure 3 legend. Stress that the measurements in A are from the glomerular layer, unlike the vessel image on the right which is pial.

7) Figure 6. Add labels for Olfactory bulb and Cortex to the panels so the reader can grasp what is plotted at a glance.

8) In the first paragraph of the subsection “Laminar organisation of capillary blood flow and Po_2_ values in the superficial layers of the somatosensory cortex” state explicitly how deep capillaries "deeper than 410 um" could be.

*Reviewer #2:*

The paper reports micron scale mapping of oxygen concentration, hematocrit, and RBC flux in both awake and anaesthetized mice using the two-photon phosphorescence probe PtP-C343. Additionally, these measurements are replicated in both the primary somatosensory cortex (n=3) as well as the olfactory bulb (n=5). Measurements of erythrocyte-associated transients were measured in individual capillaries, and oxygen tension was reported in RBCs (RBC Po_2_), in between two red blood cells (InterRBC Po_2_) and the mean between these two values (Mean Po_2_). These measurements are report for each layer of the cortex.

This manuscript provides an impressively thorough analysis of the correlation between hematocrit, RBC flux and the three oxygen concentration measurements in the bloodstream. Additionally, the comparison between awake and anaesthetized mice clearly demonstrates the increased oxygen concentration in the bloodstream. Additionally, inclusion of a comparative study with previous measurements of bloodstream oxygen concentration in both the awake and anaesthetized mice would strengthen the argument that isoflurane increases interstitial Po_2_. The explanation of how isoflurane increases interstitial Po_2_ is left somewhat unclear.

The reviewers recommend that this manuscript be published with minor revisions.

Technical Comments:

1) A more thorough clarification of the argument of how isoflurane increases interstitial oxygen concentration would strengthen the manuscript. The current explanation links increased bloodstream oxygenation to the vasodilatory effect of isoflurane (subsection “Brain oxygenation is greatly enhanced by isoflurane anesthesia”). However, the increase in RBC flux (Figure 3) resulting from vasodilation would lead to a decrease in oxygen unloading from RBCs, as evidenced by the increase in RBC Po_2_ (Figure 3). If less oxygen is unloaded from the red blood cells, then less oxygen diffuses through the vessel wall, leading to a decrease in tissue oxygenation. This is contradicted by the statement that cellular processes occur at a higher oxygen concentration during isoflurane anaesthesia. This matter is further confused by the delivery of isoflurane in a 21% oxygen gas mixture.

2) The authors repeatedly stress the importance of differentiating between RBC Po_2_, InterRBC Po_2_, and Mean Po_2_ – stating that previous measurements using two-photon phosphorescence only report Mean Po_2_. Could the authors please include a comparison of their Mean Po_2_ values against those made by other groups also using two-photon phosphorescence probe?

3) Similarly, the authors cite previous low-resolution studies on the effects of isoflurane on brain tissue oxygenation, but the manuscript is lacking a side by side comparison with previous studies. An inclusion of the previous measurements would strengthen the conclusion made in this paper.

4) Theoretical models are cited as evidence that InterRBC Po_2_ is in equilibrium with the surrounding tissue (subsection “Capillary flow and Po_2_ values in the somatosensory cortex of the awake resting mouse”). The theoretical models rely on accurate hematocrit prediction to determine the distribution of oxygen to the surrounding tissue. The relationship between radial oxygen gradients around capillaries, hematocrit and RBC velocities have been discussed in (I. Gould, A. Linninger, "Hematocrit Distribution and Tissue Oxygenation in Large Microcirculatory Networks", Microcirculation, 22, p1-18, 2015). The authors may consider inclusion of this reference to support the argument.

*Reviewer #3:*

The authors describe methods and results using 2-photon phosphorescence lifetime mediated measurements of tissue oxygenation in vivo. The authors extend their exciting work demonstrating that local O_2_ tension changes over the time course of each RBC passing (up to 100 pass per second, with each making a local O_2_ transient). This work extends a proven strategy of using compounds with a phosphorescence lifetime that is oxygen sensitive. The approach enables probe-specific calibration of oxygen levels using the phosphorescence lifetime. The methods (O_2_ probe) although not developed by Charpak`s lab, is applied quite effectively here. The authors extend previous work by David Boas and make measurements of oxygenation from awake animals. The work examines different cortical layers and contrasts with olfactory bulb. The analysis and presentation of the data is strong. Novel data includes capillary specific measurements and relationships between local oxygenation and red blood cell flux. The work also is significant for examining potentially confounding effects of well-used anaesthesia such as isoflurane. In this way the current work strikes an important cautionary tone with respect to anesthesia.

1) The authors’ analysis is impressive and I don`t have any real concerns with it, since I am familiar with this literature. However, for those which have seen these phosphorescence measurements for the first time, I think it would be important to show supplementary data illustrating how the measurements are obtained. This could be the calibration curve or some raw data showing how Po_2_ and velocity/flux are calculated in a single experiment.

2) I would also like to see the authors give some measurements for arteriole and vein diameters with isoflurane anaesthesia vs. the control awake state, the example images point towards a robust finding.

3) The authors mention that they anaesthetise all mice before initiating even awake imaging experiments. Is it possible to head fix the mice without using anaesthesia? Is the anaesthesia necessary to do the retro-orbital injection of the fluorescein dextran and oxygen probe? It seems awake mice could be tail/vein injected. This might also affect the results since there could residual effects of anaesthesia. At the very least this confound should be acknowledged.

4) The authors should comment on how their results relate to recent work from David Boas lab which propose that significant exchange of oxygen with tissues can occur through veins.

5) The authors should also comment on whether there are changes on efficacy of O_2_ signal overtime. From the protocol it sounds as though this probe must have an extremely long half-life. Why this probe would not be cleared by the liver is uncertain, its pharmacokinetics should be further investigated or at least if there is information this should be discussed so others can have a better feel for how long after injection can quantification of O_2_ be done.

6) I thought the idea that some regions of cortex could have very low flow and oxygenation to be interesting, however this is also an acute preparation and one needs to be certain that this is not due to some small amount of tissue damage when making the imaging window.

---

## [Author Response]

The reviewers have discussed the reviews with one another and the Reviewing editor has drafted this decision to help you prepare a revised submission. All reviewers are enthusiastic about this work. Each raised a number of questions. I ask you address each question by clarifying the manuscript. No new data is required. I have summarized the key questions:

*1) The hematocrit is reported to be significantly higher in cortex than in olfactory bulb (subsection “Capillary flow and Po_2_ values in the somatosensory cortex of the awake resting mouse”). This needs to be discussed* –

*how is this difference produced? Is it somehow produced by plasma skimming at a series of junctions of the vessels supplying the olfactory bulb? As this is an acute preparation, one needs to be certain that this is not due to some small amount of tissue damage when making the imaging window.*

The fact that cortical capillary hematocrit is higher than that measured in the olfactory bulb glomerular layer is indeed an interesting finding. However, the mechanisms that underlie this difference are not easy to determine. We agree that differential hemoconcentration or hemodilution in vessels supplying the recorded capillaries could account for some of this difference. A modelling approach, such as that developed by Gould and Linninger (2015), could provide insight into this issue, however a definitive answer to this question remains elusive, as the details of the blood supply to the olfactory bulb are unknown. This is a weakness of the model, and we are just beginning a collaboration to investigate the anatomical organisation of the bulb vasculature. We hope to be able to answer this question in the future. We have added a short note on this point in the first paragraph of the subsection “Capillary flow and Po2 values in the somatosensory cortex of the awake resting mouse“.

Concerning the second point:

Conscious of the fact that a chronic cranial window preparation might affect our Po_2_ measurements, when starting this project, we initially employed a chronic thinned-skull window preparation (Drew et al. 2010), which avoided opening the cranium. However, after a number of months using this technique, we determined that the phosphorescence decays detected and the data recorded in this preparation were not compatible with high quality recordings, so we switched to using implanted glass cranial windows. Nonetheless, the facts that: 1. Our approach of placing the cover glass into the craniotomy and directly on to the dura mater surface seems to avoid the commonly described degradation (due to inflammation) and subsequent recovery of cranial window quality over the course of 2-3 weeks (Holtmaat et al. 2009), and 2. Our measurements were made many days after the chronic cranial window surgery suggest that we largely avoided damage to the tissue that would cause our results to deviate significantly from normal physiology.

2) Please show a raw trace of the Po_2_

*profile between RBCs, or state explicitly that the line in the inset to panel A IS raw data (and not a schematic); why is that line continuous to the left and dotted on the right hand side?*

The solid black trace in the lower right panel of Figure 1 was indeed data, drawn from recordings in anaesthetised mice as reported in Parpaleix et al. (2013). It has now however been replaced by a trace representing data from one capillary recorded in the awake mouse for this study. The continuous line represents the data trace, showing the profile of Po_2_ relative to the border of an RBC. The dotted portion of the trace is a mirror image of the data trace, and is used in the schematic solely for the purposes of illustration. The figure legend has been amended accordingly.

*3) The authors repeatedly stress that previous measurements using two-photon phosphorescence only report Mean Po_2_. Could the authors please include a comparison of their Mean Po_2_ values against those made by other groups also using two-photon phosphorescence probe?*

A paragraph has been added to the Discussion (second paragraph) to compare our recorded values of Po_2_ to those previously made with 2PLM by other groups.

*4) Please explain the bars in*
Figure 1 far better. State completely clearly that *Po_2_ is the mean (over vessels) of a single time averaged value obtained from each vessel, while SD is mean (over vessels) of the standard deviation of the Po_2_*

*in each vessel over time (if that is what it is!).*

The necessary clarifications have been made in the legend for Figure 1.

5) The Po_2_ seems (subsection “Laminar organisation of capillary blood flow and Po_2_

*values in the superficial layers of the somatosensory cortex”) to be determined by the local capillary density. Does this imply that local O_2_ consumption is independent of laminar position (one might have expected vessel density to be proportional to local O_2_ usage)? Is it possible to derive, from the laminar distribution of [O_2_] and the blood flow reaching each lamina, how much O_2_ is consumed by each cell layer?*

Although our findings show that, at rest, cortical laminar Po_2_ is correlated to local capillary density, these data do not demonstrate any causal relationship, nor do they imply that the local rate of oxygen consumption is independent of the vessel density. Our lab is beginning to investigate the respective importance of cellular activity and delivery of oxygen in regulating tissue Po_2_ per layer. We plan to examine the effects of blocking pre/post-synaptic activity and firing on local tissue Po_2_. This research will help to clarify the factors that determine laminar and regional Po_2_.

We have amended the text of the manuscript to remove the implication of a causal relationship between laminar Po_2_ and capillary density (subsection “Laminar organisation of capillary blood flow and Po_2_ values in the superficial layers of the somatosensory cortex”, second paragraph).

6) A more thorough clarification of the argument of how isoflurane increases interstitial oxygen concentration would strengthen the manuscript. The current explanation links increased bloodstream oxygenation to the vasodilatory effect of isoflurane (subsection “Brain oxygenation is greatly enhanced by isoflurane asesthesia”). However, the increase in RBC flux (Figure 3) resulting from vasodilation would lead to a decrease in oxygen unloading from RBCs, as evidenced by the increase in RBC Po_2_

*(Figure 3). If less oxygen is unloaded from the red blood cells, then less oxygen diffuses through the vessel wall, leading to a decrease in tissue oxygenation. This is contradicted by the statement that cellular processes occur at a higher oxygen concentration during isoflurane anaesthesia. This matter is further confused by the delivery of isoflurane in a 21% oxygen gas mixture.*

It is true that increased RBC velocity (mm s^-1^) will lead to a decrease in oxygen unloading from each RBC. However, the overall increase in the RBC flux (cells s^-1^) that we measured will mean that although less oxygen is unloaded from each individual RBC, the total quantity of oxygen diffusing through the vessel wall could increase, leading to an increase in tissue oxygenation. This can be thought of as being somewhat similar to the increase in both RBC hemoglobin saturation and tissue oxygenation that is observed in a glomerulus during functional hyperaemia (Figure 4, Parpaleix et al. 2013).

Although isoflurane is commonly delivered by others in either 100% oxygen or an air-oxygen mixture (where [O_2_]>21%), in our study we delivered it in air without additional oxygen ([O_2_]≈21%, (Line 142-143 of the track-changes version of manuscript). Our statement that “this difference is likely to be exacerbated when isoflurane is delivered in gas mixtures where [O_2_] is greater than 21%” was not made in reference to our study, but rather to the delivery of isoflurane in an air-oxygen mix by other researchers. This is now better clarified in the text.

*7) Theoretical models are cited as evidence that InterRBC Po_2_ is in equilibrium with the surrounding tissue (subsection “Capillary flow and Po_2_*

*values in the somatosensory cortex of the awake resting mouse”). The theoretical models rely on accurate hematocrit prediction to determine the distribution of oxygen to the surrounding tissue. The relationship between radial oxygen gradients around capillaries, hematocrit and RBC velocities have been discussed in (I. Gould, A. Linninger, "Hematocrit Distribution and Tissue Oxygenation in Large Microcirculatory Networks", Microcirculation, 22, p1-18, 2015). The authors may consider inclusion of this reference to support the argument.*

We thank the reviewer for this advice, and have cited this paper accordingly (subsection “Capillary flow and Po_2_ values in the somatosensory cortex of the awake resting mouse”, first paragraph).

*8) The authors mention that they anaesthetise all mice before initiating even awake imaging experiments. Is it possible to head fix the mice without using anaesthesia? Is the anaesthesia necessary to do the retro-orbital injection of the fluorescein dextran and oxygen probe? It seems awake mice could be tail/vein injected. This might also affect the results since there could residual effects of anaesthesia. At the very least this confound should be acknowledged.*

For awake recording sessions in our trained animals, we could head-fix the mice without using anaesthesia and without generating any stress. Technically, we preferred using a brief anaesthesia (< 5minutes) to perform a rapid retro-orbital injection rather than the tail/vein injection. However, to avoid any potential confound related to this brief exposure to isoflurane (2%) we:

A) Ensured a delay of ~2 hours between the period of isoflurane exposure and the initiation of the first set of recordings;

B) Performed multiple sessions of recordings over 1-2 days without reanaesthetising and injecting another bolus of PtP-C343 and fluorescein dextran. The data gathered in each of these sessions was comparable, suggesting that even the first set of recordings after the injection procedure was not affected by the exposure to isoflurane.

Figure 8 shows Po_2_ values recorded from a single cortical capillary during 3 separate recording sessions across 2 days following the retro-orbital injection, demonstrating that Po_2_ values from this capillary were similar at ~2 hours (Session 1), ~7 hours (Session 2) and ~24 hours (Session 3) after the initial 5 minute exposure to isoflurane.

Author response image 1.**DOI:**
http://dx.doi.org/10.7554/eLife.12024.011

We have amended the text of the paper to better reflect our procedures for injection of PtP-C343, head-fixation and awake recordings (subsection “Habituation of mice to head restraint“).

*Reviewer #1:*

[…]

*3) It is surprising that this study finds a lower Po_2_*

*in the surface layers of the cortex, whereas Devor and Sakadzic found a decrease in [O_2_] with depth. Is there any methodological difference that could account for this, e.g. O_2_ penetration into the brain from the cortical window in the earlier studies)? Was the laminar profile of O2 affected by isoflurane?*

Sakadžić et al. (2010, 2014), and Devor (2011) sealed their cranial windows before their Po_2_ recordings, thus limiting the likelihood of oxygen diffusion into the brain directly from the atmosphere during the experiment. It is thus most likely that the decrease in Po_2_ with depth they reported is related to either the effects of anaesthesia, acute surgical procedures, or indeed a combination of these factors.

Our goal was mostly to stress the global effects of isoflurane. We thus unfortunately did not assess the effect of isoflurane on the laminar Po_2_ profile in the cortex, limiting paired measurements in capillaries from layer I.

[…]

6) Figure 3 legend. Stress that the measurements in A are from the glomerular layer, unlike the vessel image on the right which is pial.

The necessary alteration has been made to the figure legend.

[…]

*8) In the first paragraph of the subsection “Laminar organisation of capillary blood flow and Po_2_ values in the superficial layers of the somatosensory cortex” state explicitly how deep capillaries "deeper than 410 um" could be.*

410 µm was the maximum depth at which we made measurements, and this fact has now been clarified in the manuscript (subsection “Laminar organisation of capillary blood flow and Po2 values in the superficial layers of the somatosensory cortex”, first paragraph).

*Reviewer #2:*

Technical Comments:

[…]

*2) Similarly, the authors cite previous low-resolution studies on the effects of isoflurane on brain tissue oxygenation, but the manuscript is lacking a side by side comparison with previous studies. An inclusion of the previous measurements would strengthen the conclusion made in this paper.*

Rather than present a side by side comparison of our data with that of previous studies (as in a table), we present a significant discussion of the data from these lower resolution studies (citing their reported values of Po_2_. Discussion, first paragraph).

Reviewer #3: 1) The authors’ analysis is impressive and I don`t have any real concerns with it, since I am familiar with this literature. However, for those which have seen these phosphorescence measurements for the first time, I think it would be important to show supplementary data illustrating how the measurements are obtained. This could be the calibration curve or some raw data showing how Po_2_

*and velocity/flux are calculated in a single experiment.*

Rather than showing supplementary data, we now explicitly refer the reader to figures from both Parpaleix et al. (2013) and Lecoq et al. (2011), which present the measurement techniques we employ in great detail (subsection “Capillary blood flow analysis and EAT properties extraction”, second paragraph).

*2) I would also like to see the authors give some measurements for arteriole and vein diameters with isoflurane anaesthesia vs. the control awake state, the example images point towards a robust finding.*

Although we are able to show example images, and noticed the obvious dilation of pial vessels during isoflurane experiments, regretfully we did not systematically collect paired data on vessel diameter and cannot give serious statistics.

*5) The authors should also comment on whether there are changes on efficacy of O_2_ signal overtime. From the protocol it sounds as though this probe must have an extremely long half-life. Why this probe would not be cleared by the liver is uncertain, its pharmacokinetics should be further investigated or at least if there is information this should be discussed so others can have a better feel for how long after injection can quantification of O_2_ be done.*

See our response to the Key question 8. Further information on the interval following PtP-C343 injection during which accurate Po_2_ measurements can be made has been added to the manuscript (subsection “Habituation of mice to head restraint”, second paragraph).